# Biomass Calculations of Individual Trees Based on Unmanned Aerial Vehicle Multispectral Imagery and Laser Scanning Combined with Terrestrial Laser Scanning in Complex Stands

**Xugang Lian** [1,*] , **Hailang Zhang** [1] , **Wu Xiao** [2] , **Yunping Lei** [3] , **Linlin Ge** [1,4] , **Kai Qin** [5] , **Yuanwen He** [1] , **Quanyi Dong** [1] , **Longfei Li** [1] , **Yu Han** [1] , **Haodi Fan** [1] , **Yu Li** [1] , **Lifan Shi** [1] and **Jiang Chang** [1]

1   Department of Geomatics, Taiyuan University of Technology, Taiyuan 030024, China
2   Department of Land Management, Zhejiang University, Hangzhou 310058, China
3   Land Space Surveying and Planning Center of Shanxi Province, Taiyuan 030021, China
4   School of Civil and Environmental Engineering, UNSW Australia, Sydney, NSW 2052, Australia
5   School of Environment Science and Spatial Informatics, China University of Mining and Technology, Xuzhou 221116, China
*   Correspondence: lianxugang@tyut.edu.cn

**Abstract:** Biomass is important in monitoring global carbon storage and the carbon cycle, which quickly and accurately estimates forest biomass. Precision forestry and forest modeling place high requirements on obtaining the individual parameters of various tree species in complex stands, and studies have included both the overall stand and individual trees. Most of the existing literature focuses on calculating the individual tree species' biomass in a single stand, and there is little research on calculating the individual tree biomass in complex stands. This paper calculates the individual tree biomass of various tree species in complex stands by combining multispectral and light detection and ranging (LIDAR) data. The main research steps are as follows. First, tree species are classified through multispectral data combined with field investigations. Second, multispectral classification data are combined with LIDAR point cloud data to classify point cloud tree species. Finally, the divided point cloud tree species are used to compare the diameter at breast height (DBH) and height of each tree species to calculate the individual tree biomass and classify the overall stand and individual measurements. The results show that under suitable conditions, it is feasible to identify tree species through multispectral classification and calculate the individual tree biomass of each species in conjunction with point-cloud data. The overall accuracy of identifying tree species in multispectral classification is 52%. Comparing the DBH of the classified tree species after terrestrial laser scanning (TLS) and unmanned aerial vehicle laser scanning (UAV-LS) to give UAV-LS+TLS, the concordance correlation coefficient (CCC) is 0.87 and the root-mean-square error (RMSE) is 10.45. The CCC and RMSE are 0.92 and 1.41 compared with the tree height after UAV-LS and UAV-LS+TLS.

**Keywords:** tree species identification; DBH; tree height; biomass calculation

## 1. Introduction

Increased carbon dioxide emissions have led to a gradual warming of the global climate and a sharp deterioration of the ecological environment. Photosynthesis in terrestrial ecosystems can effectively relieve carbon dioxide emissions [1,2]. As the largest carbon storage pool on land, forests effectively control carbon dioxide emissions, regulate the global carbon cycle, and slow down climate warming. As an important indicator of vegetation life activities, forest biomass is important in monitoring global carbon storage and the carbon cycle to evaluate terrestrial ecosystems. Therefore, rapid and accurate estimations of forest biomass are significant for forest ecological management systems and climate decision support [3,4].

Previous researchers have mainly obtained the DBH and tree height through field investigations. The DBH is easily obtained, but the tree height is not only difficult to obtain but is also vulnerable to various errors (plant factors, topographic factors, human errors, instrument errors, etc.). Thus, inventorying forest resources is time-consuming, laborious, and subjective [5–8].

The development of optical remote sensing technologies has brought convenience to the inventory of forest resources and has advantages in monitoring the dynamics of forest biomass. Researchers have widely applied multispectral or hyperspectral imaging to classify forest tree species, identify biological species based on the spectral information and reflectance of plants, and establish a statistical relationship model between the vegetation index and biomass at different growth stages through the normalized vegetation index (NDVI) to calculate forest biomass. As images can only provide 2D data, they cannot provide the vertical structure information of trees; thus, research is more focused on the stand level with the average value or sum of stands obtained [9–11]. Precision in forestry, forestry management planning, and forestry growth modeling plays increasingly important roles in forestry applications, so more attention has been given to the acquisition of individual tree parameters (tree height and DBH) of various tree species in stands. Research starts from the overall stand to the study of individual trees. The acquisition of individual tree parameters (tree height and DBH) through automatic individual tree detection (ITD) is also becoming increasingly important in forestry applications [12].

With improvements in dense image-matching methods and computer capabilities, digital aerial photography (DAP) measurements can generate point cloud data through the structure-from-motion (SFM) algorithm, which contains a significant amount of tree-top information. The digital terrain model (DTM) data can be subtracted from the digital surface model (DSM) data to calculate the canopy height model (CHM), and individual trees can be segmented to obtain the individual tree parameters [13–16]. However, accurate ground information cannot be provided in complex stands, and the resulting forest structure information has a high uncertainty [17,18].

With the miniaturization and low cost of the Global Navigation Satellite System (GNSS) and inertial navigation systems, unmanned aerial vehicles (UAVs) are becoming more popular in forestry research because of their flexible route planning, low cost, reliability and convenience, and ability to quickly provide high-resolution data [19–23]. Unmanned aerial vehicles carrying multispectral sensors can classify the spectral characteristics of diverse forest areas in forestry, obtain the spatial distribution of tree species, and classify statistics of areas of tree species. By multi-spectral imaging and automatic model extraction, we can extract the target of diseased trees, assess the growth status of trees through the spectral reflection characteristics of trees, quantitatively analyze the indicators of forests, and enrich the results of forestry monitoring [24,25]. Researchers have introduced laser scanning (LS) equipment (such as terrestrial laser scanning (TLS) and unmanned aerial vehicle laser scanning (UAV-LS)) into ground and air platforms. LIDAR remote sensing has great advantages when calculating forest biomass because of its high resolution and rapid acquisition of forest vertical structural parameters [26–28]. With its high penetration, UAV-LS can efficiently obtain scale data above the stand, such as the crown area and height information of individual trees. However, in locations with complex stand structures, especially in areas with many twigs, dense stands, and young seedlings, the model under the crown has high uncertainty [29–31]. Compared with UAV-LS, TLS relies on its high scanning accuracy and scanning density to obtain accurate DBH of individual trees. However, it is not easy to obtain the crown area above the trees when collecting data. researchers combined UAV-LS and TLS data to carry out the retrieval of forest parameters such as the tree crown height model, generated from a normalized point cloud [32], seed point individual tree segmentation [33], and QSM modeling [31]. The combination of UAV-LS and TLS can greatly improve the extraction of individual tree parameters and provide reasonable prediction errors for volume calculations [34,35].

Most of the existing literature focuses on calculating the individual tree biomass of species in a single stand, and there is less research on this topic for complex stands. When acquiring individual tree parameters in complex stands, optical remote sensing can divide the stands but lacks three-dimensional (3D) data for individual trees. LIDAR can provide 3D data of individual trees but it cannot identify tree species [36–38]. Combining the data of multispectral forest division with LIDAR point cloud data allows identifying each tree species in a forest to obtain the parameters of individual tree biomass. Given these shortcomings, this study combines multispectral, UAV-LS, TLS, and other multi-source remote sensing data through the spectral identification of tree species: UAV-LS to obtain tree height and TLS to obtain DBH.

A practical UAV-LS+TLS+ multispectral method is proposed to estimate forest biomass. This method is applied to the Huyu campus of Taiyuan University of Technology. The tree species in the research area are classified by multispectral and field investigations, and the classified data are combined with LIDAR data to classify the tree species of the point-cloud data. The biomass of the classified point cloud tree species is then calculated. This study helps estimate forest biomass in a more convenient, efficient, and low-cost way compared to conventional approaches.

## 2. Materials and Methods

### 2.1. Study Site

A park in the Huyu campus of Taiyuan University of Technology was selected as the research area, which is located in the middle of Shanxi Province and the northern part of Jinzhong Basin. The geographical coordinates are 37°51′10″ to 37°51′12″ North and 112°31′8″ to 112°31′12″ East. The highest altitude in the study area is 799 m and the lowest altitude is 786 m; the average is 792 m. The area of the observed site is 0.007 km$^2$, and 115 trees were surveyed. The main tree species in the area are poplar, willow, lacebark pine, cypress, and clove tree. The location of the study area is illustrated in Figure 1.

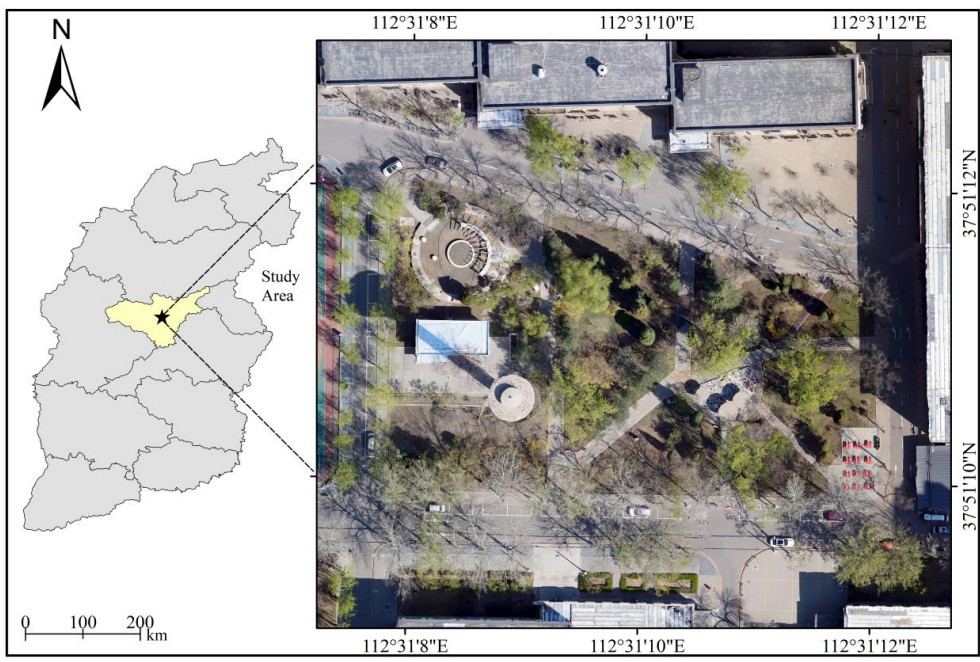

**Figure 1.** Location of the study area as marked by a star.

### 2.2. Field Data Collection

#### 2.2.1. Manual Data Collection

The species were investigated at the chosen location in April 2022. In April, trees in the study area had just sprouted and the collection of tree characteristic data at this time had

little impact, which is beneficial for later vegetation identification and vegetation biomass modeling calculations. Consequently, we collected our data in April. The morphological characteristics of each species were considered, the position coordinates of the trees were measured with the GNSS instrument, and the diameter at the breast height of each tree was measured with a leather ruler. The GNSS equipment used is Hi-Target RTK, whose plane accuracy is $\pm 0.25$ m + 1 ppm and elevation accuracy is $\pm 0.50$ m + 1 ppm. Table 1 describes the tree species and morphological characteristics.

**Table 1.** Tree species and morphological characteristics from the study area.

| Tree Species | Trees | Morphological Characteristics |
|---|---|---|
| Willow | 17 | Switch branches are slender, soft, and drooping; bark tissue is thick and longitudinally split; and the center of the old trunk is rotten and hollow. |
| Poplar | 12 | Bark grayish brown, fissured at the lower part; and sprouts thin, round, smooth, or slightly tomentose. |
| Clove tree | 42 | The trunk is forked; the crown is conical; and the bark is smooth, yellowish brown. |
| Lacebark pine | 15 | There is a large trunk; the branches are slender, obliquely spread, tower-shaped or umbrella-shaped crown; and winter buds are reddish-brown, oval, without resin. |
| Cypress | 29 | The bark is dark gray, and young trees often have branches that extend diagonally to form a spire-shaped canopy. |

### 2.2.2. LIDAR Remote Sensing Data

(1)  Terrestrial Laser Scanning (TLS) Data

In April 2022, the data were gathered using Leica total station scanner MS50, with a ranging accuracy of 0.6 mm. MS50 is a total station scanner that integrates 3D laser scanning technology, high-precision measurement technology, digital imaging technology, and GNSS technology [39]. The scanning speed is 1000 points/s within 300 m. MS50 is a total station scanner that integrates 3D laser scanning technology, high-precision measurement technology, digital imaging technology, and GNSS technology. A total of 21 stations were set up, and in order to ensure the same resolution of scanning results between different stations, the same point cloud density parameters were set before each station started scanning. When the scanning horizontal distance was set to 15 m, the obtained scanning point had a horizontal distance of 10 cm and a vertical distance of 10 cm. The scanning speed was 1000 points per second, the estimated number of points was about 40 thousand, and the scanning took 25 min. The parameters of MS50 are shown in Table 2.

**Table 2.** MS50 parameters.

| MS50 | | |
|---|---|---|
| | **Prism (GPR1,GPH1P)** | 1.5 to 3500 m |
| Range | No prism/any surface | 1.5 m to >1000 m |
| | Reflector (60 mm × 60 mm) | 250 m |
| Accuracy/ Measurement | single time (prism) | 0.6 mm + 1 ppm/ typically 2.4 |
| | Single (any surface) | 2 mm + 2 ppm/typically 3 s |
| Spot size | At 50m | 8 mm × 20 mm |
| Measurement technology | System analysis technology based on phase principle | Coaxial, red visible light |

(2)  Unmanned aerial vehicle laser scanning (UAV-LS) data

The UAV-LS data were collected using the FEIMA robotics D-LIDAR in April 2022. The flight height of the aircraft was 128 m and the flight speed was 13.5 m/s. The average point density was $286/m^2$, and up to three echoes were obtained with an echo intensity of 8 bits. The laser pulse wavelength and frequency were 905 nm and 240 kHz. The collected data were processed by the UAV Manager software, including the GNSS and inertial measurement unit (IMU), and the accuracy after processing was such that the position

accuracy and the attitude accuracy were not more than 0.02 m and 3°, respectively. Table 3 provides the parameters of LIDAR data acquisition.

**Table 3.** Parameters of LIDAR data acquisition.

| FEIMA D-LIDAR 2000 Module | | | |
|---|---|---|---|
| **Laser Type** | **RIEGL mini VUX-1UAV** | **Channels** | 1 |
| Dot Frequency | 100 kpts/s | Measurement Range | 250 m |
| Range Accuracy | ±1 cm | Echo Number | 5 (Max.) |
| Scanning Speed | 10~100 Hz | Echo Intensity | 16 bit |
| Wavelength | 905 nm (Class 1) | Laser Divergence Angle | $1.6 \times 0.5$ mrad |
| Horizontal Field of View | 360° | Resolution-horizontal | 0.05~0.5° |

### 2.2.3. Optical Remote Sensing Data

(1) Orthophoto Data

During the collection in April 2022, the weather was sunny and the sky was cloudless. We collected the aerial images using the D-CAM2000 mounted on the FEIMA robotics UAV. The side heading overlap is 80%, the overlap is 60%, and the ground sampling distance (GSD) is 2.0 cm. The flight height of the aircraft was 100 m, and the flight speed was 13.5 m/s. The data were processed using the UAV Manager software, and the accuracy after processing was 2 cm. Table 4 presents the parameters of orthophoto data acquisition.

**Table 4.** Parameters of orthophoto data acquisition.

| FEIMA robotics D-CAM2000 Aerial Survey Module | | | |
|---|---|---|---|
| Camera Type | SONY ILCE-6000 ($\alpha$6000) | Sensor Size | $23.5 \times 15.6$ mm |
| Effective Size | ($6000 \times 4000$) 2400 million | Lens | 20 mm fixed focus |
| Gimbal | 2-axis | | |

(2) Multispectral Data

In April 2022, the multispectral data of the UAV were obtained using the FEIMA robotics UAV D-MSPC2000. The spectral bands were blue (450 nm), green (555 nm), red (660 nm), infrared 1 (720 nm), infrared 2 (750 nm), and near-infrared (840 nm). The side heading overlap was 75%, the overlap was 75%, and the GSD was 7.2 cm. The flight height of the aircraft was 100 m, and the flight speed was 12.4 m/s. The collected data were processed using the Pix4D software from Switzerland and the UAV Manager software from the FEIMA company, and the accuracy after processing was 7.8 cm. The parameters of multi-spectral data acquisition are displayed in Table 5.

**Table 5.** Parameters of multi-spectral data acquisition.

| FEIMA Robotics D-MSPC2000 Multi-Spectral Module | | | |
|---|---|---|---|
| Sensor parameters | CMOS: 1/3″ global shutter | Effective pixels | 1.2 million |
| Resolution | $1280 \times 960$ | Sensor size | 4.8 mm $\times$ 3.6 mm |
| Focal length | 5.2 mm | Field of view | HFOV: 49.6°, VFOV: 38° |
| Aperture | F/2.2 | Quantization bits | 12 bit |
| Shooting speed | 1 time/s | Ground resolution | GSD: 8.65 cm/pix, AGL: 120 m |

The data acquisition is shown in Figure 2.

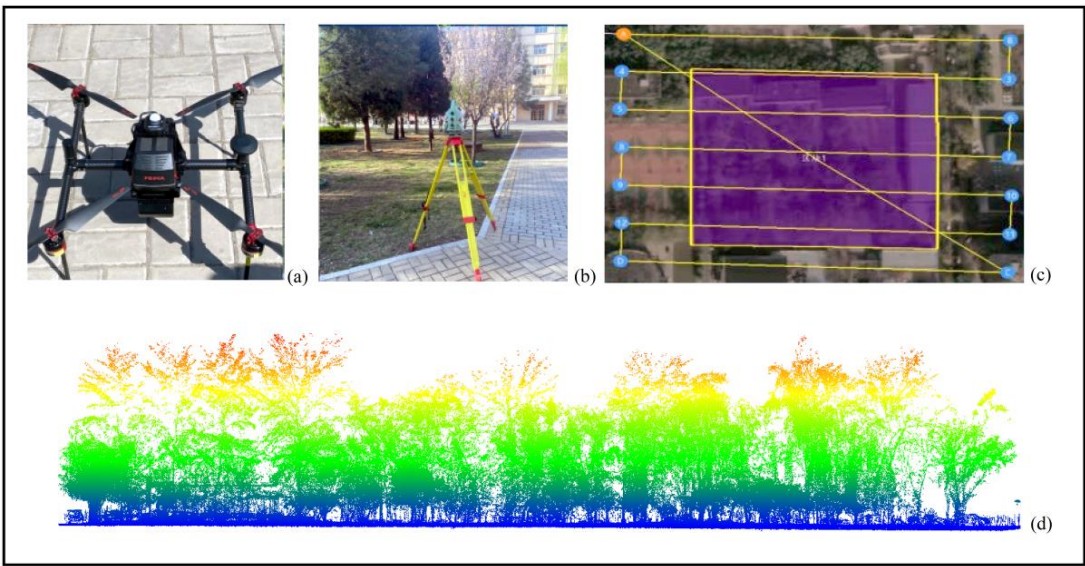

**Figure 2.** (**a**) The UAV, (**b**) The total station with the scanning mode, (**c**) The flight plan and (**d**) Point cloud image.

### 2.3. Research Methods

We first used multispectral and field surveys to classify tree species, combine the classified data with LIDAR data to divide the tree species into point clouds, use the divided data to compare the DBH and tree height, and finally calculate the individual tree biomass. The specific flow of individual tree biomass calculation is shown in Figure 3.

#### 2.3.1. Tree Species Classification in the Study Area

With the increasing maturity of spectral sensing technology, image processing, and analysis software, UAVs carrying multispectral technology have made great progress in forestry applications. The spectral reflectance of plants can be collected from multispectral data, and the tree species in the study area can be distinguished from the spectral reflectance. This study combines orthophoto data and multispectral image data to classify tree species in the study area.

(1) A multinomial distribution algorithm introduced by Mosimani and James is adopted [40]:

$$n = \max_{i \in 1,2,\cdots,k} \left( (B \times P_i \times (1 - P_i))/b_i^2 \right) \tag{1}$$

where $n$ is the total number of samples; $P_i$ is the percentage of the tree category in the total tree species; $B$ is the confidence of $\Delta = 1 - \alpha/k$ ($k$ is the number of classification categories, and $\alpha$ is the expected significance level) with a chi-square test value having a degree of freedom of 1; and $b_i$ is the expected classification error ratio percentage of the class. The minimum reasonable number of samples is then calculated.

(2) Gong and Howarth studied different sampling strategies and believed that the highest classification accuracy can be obtained by collecting a pixel sample every N pixels [41]. Here, simple random sampling with an unbiased estimation of the population parameters is used.

(3) The spectral angle classification (SAM) algorithm is used to compare and classify the unknown spectral lines with the sample spectrum in N-dimensional space [42]. By comparing the angle between the reference spectrum vector and each pixel vector, it is found that a greater angle causes the pixels to be more similar to the reference spectrum.

(4) According to the field investigation of tree species, the tree spectra are extracted from the image pure pixels to establish the spectrum library, which is used to sort the tree species in the study area, as represented in Figure 4.

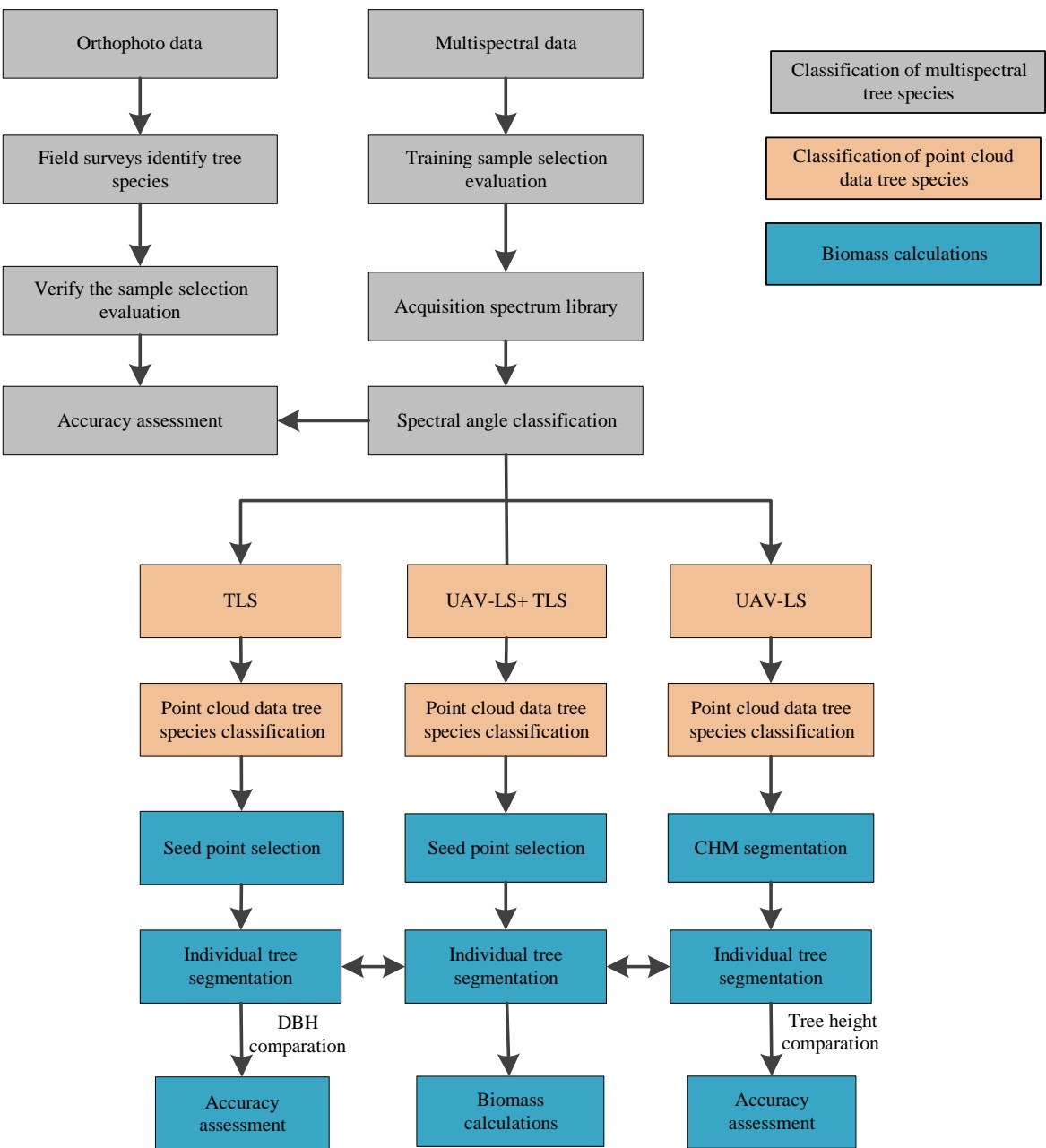

**Figure 3.** Flow of individual tree biomass calculation.

2.3.2. Individual Tree Biomass Calculation

LIDAR Point Cloud Data Tree Species Classification

(1) Fusion of UAV-LS and TLS Data

The collected UAV-LS is readily affected by the canopy and lacks information below it. The collected TLS is also readily affected by the canopy and lacks information above it. Both the UAV point cloud data and the point cloud data of the total station scanner were collected utilizing the CGCS2000 coordinate system provided by the Qianxun CORS network, and they have the same geographical reference and the same coordinate system, which can be directly fused in the software. The fusion of UAV-LS acquisition point cloud data and TLS collection point cloud data has greatly improved the accuracy of extraction of individual wood parameters and provided reasonable prediction error for volume [43,44]. Thus, the UAV-LS and TLS point clouds were integrated, as shown in Figure 5.

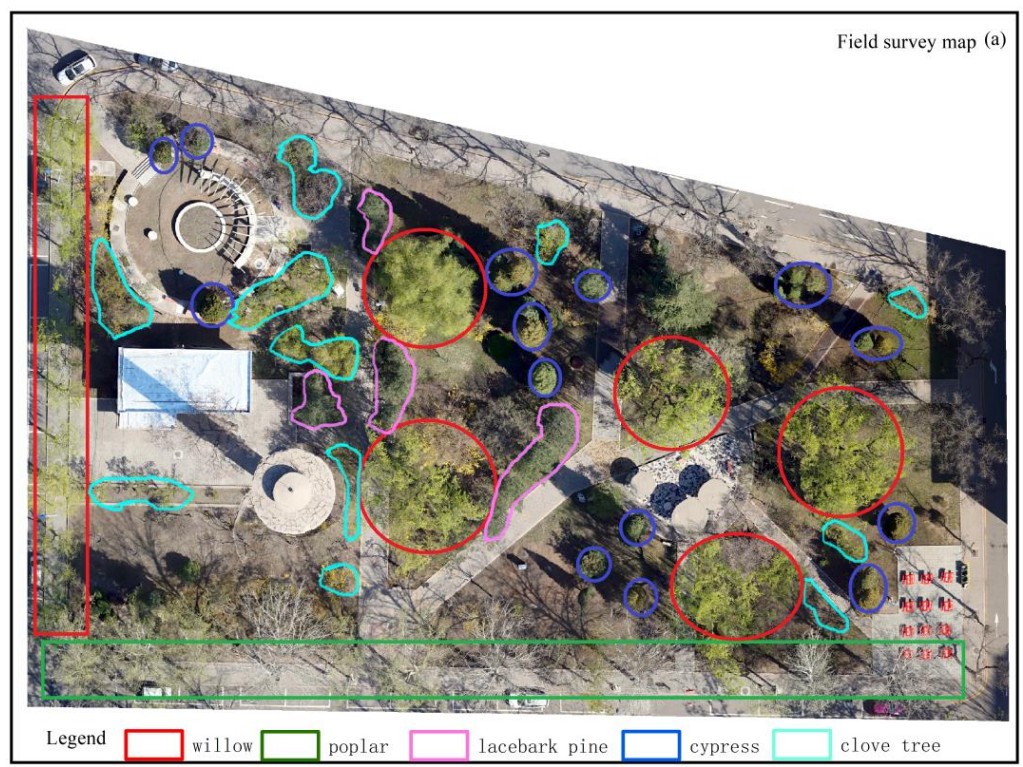

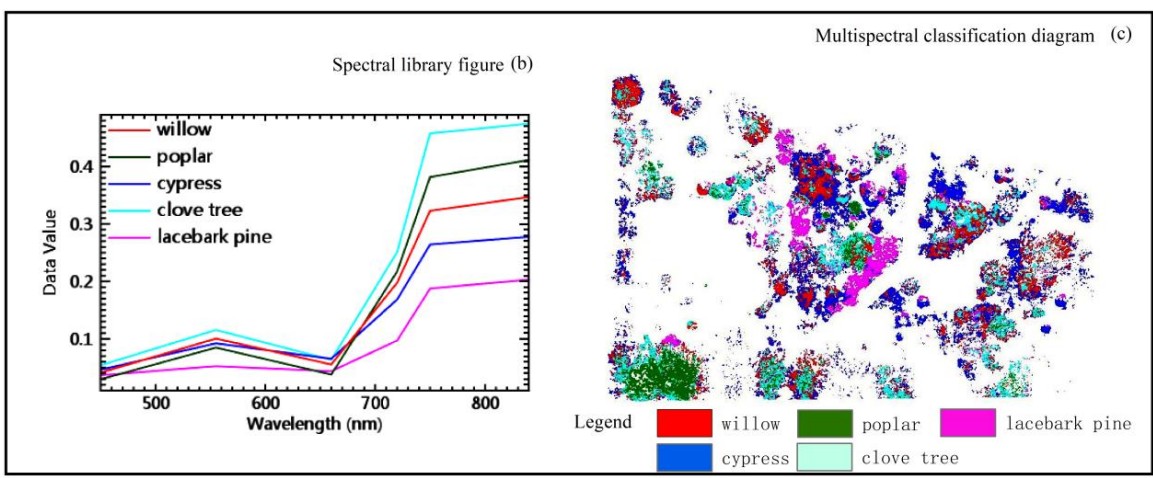

**Figure 4.** (**a**) Field survey map, (**b**) spectral library spectra, and (**c**) multispectral classification diagram.

(2)    Point cloud data for tree species classification

Multispectral data can identify and classify tree species, but only 2D data of individual trees can be obtained. LIDAR data exhibit strong penetration into plants, but they cannot classify tree species. To obtain the individual tree parameters of each tree species, it is necessary to combine multispectral classification data and LIDAR point-cloud data. The spectral angle classification result data and the LIDAR point cloud data are first loaded in the Cloud Compare software. In the case of the same coordinate system registration, the spectral angle classification result in the interface will be displayed on the top, and the LIDAR data will be on the bottom. By cutting the data, the LIDAR data will be trimmed and filtered out according to the spectra-angle classification results, and consequently, the LIDAR data of each tree species will be formed. The results of the spectral angle classification are combined with the LIDAR point cloud data to classify each tree species, as shown in Figure 6.

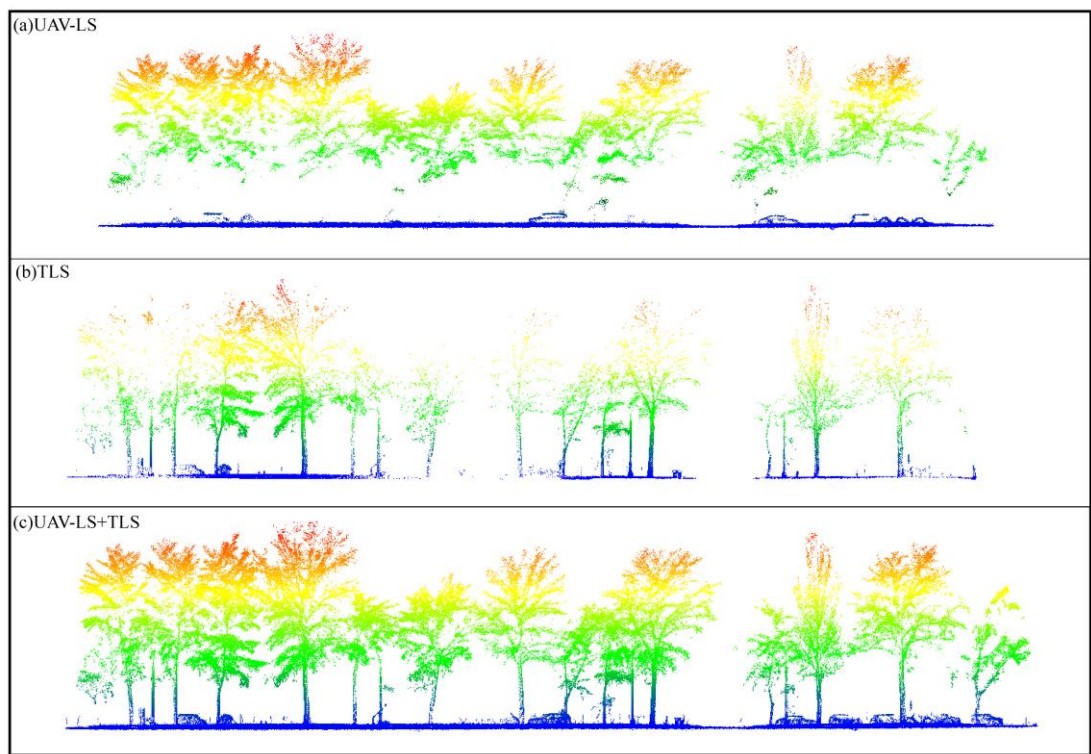

**Figure 5.** Point cloud data graph combining the TLS and UAV-TLS collections.

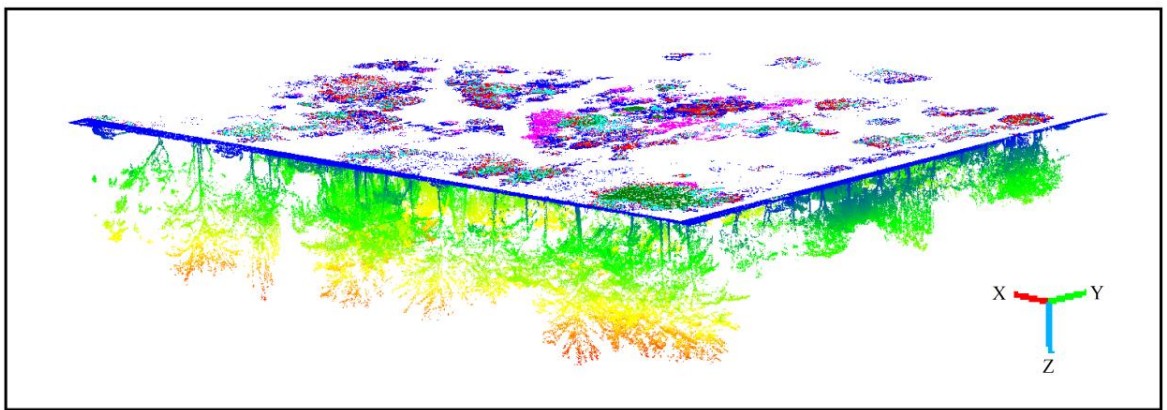

**Figure 6.** (**Top**): Multispectral classification data; (**Bottom**) Point-cloud data.

Calculation of Individual Tree Biomass of Classified Tree Species

After tree species classification, the individual tree parameters can be extracted from each classified tree. The preprocessing of point cloud data, ground point classification, individual tree segmentation, etc., is performed using the LIDAR 360 software.

(1) The noise mainly includes high-position gross error and low-position gross error. The algorithm searches adjacent points within a specified neighborhood; calculates the average distance $d$ from the point to the adjacent points; calculates the median, mean, and standard deviation $\sigma$ of the average distances; and removes noise by selecting appropriate parameters.

(2) The improved progressive triangulation filtering algorithm (iPTD) is used to classify ground points [45]. A sparse triangulation is generated through seed points and is later encrypted by the layer through iterative processing until all ground points are classified.

(3) The digital terrain model (DTM) is removed from the digital surface model (DSM) to obtain the canopy height model (CHM) [46]. Through the upstream ridge segmentation algorithm, the high points of the CHM are regarded as peaks, and the low points are regarded as valleys. The water areas are filled, and barriers are built as the water edges as determined from segmentation. In experiments, when performing CHM segmentation, a tree is often identified as several trees based on the algorithm alone, resulting in multi-division so that there are more trees after CHM segmentation than actual trees. Consequently, we need to combine CHM segmentation and manual operations to classify trees. The data classification was conducted using Cloud Compare and LIDAR360 software. The individual tree segmentation of the seed points is used to define the parameter variables such as the tree height and DBH, as shown in Figure 7.

(4) The volume calculations of the study area adopt the individual tree binary volume model volume $V = aD^bH^c$, where $a$, $b$, and $c$ are model parameters; $D$ is the DBH; and $H$ is the tree height.

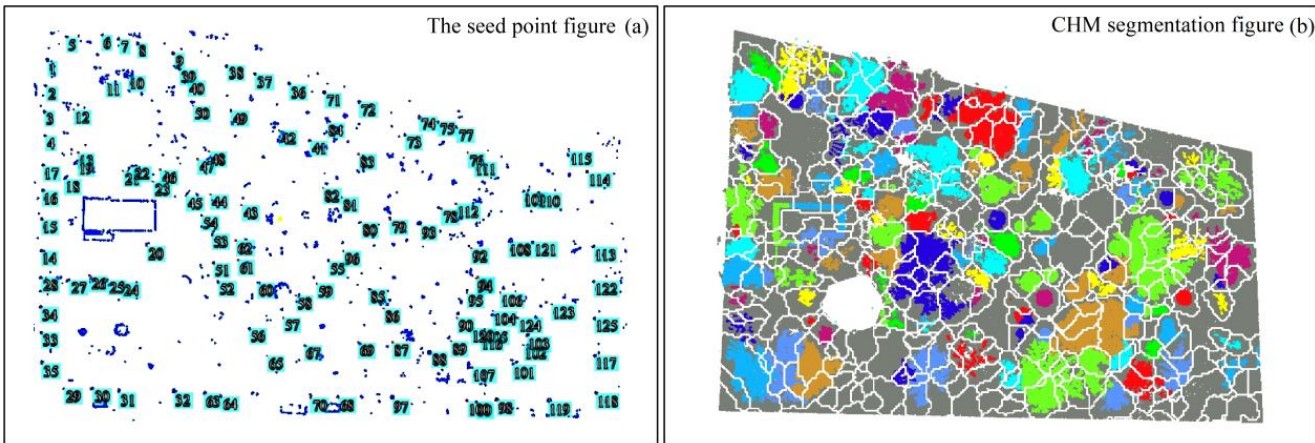

**Figure 7.** (**a**) Seed point and (**b**) CHM segmentations.

We calculate the biomass for willow, poplar, clove, and laceback pine by adopting the individual tree biomass model. The total above-ground biomass of $W_T = a(D^2H)^b$, underground biomass of $W_R = W_T/3.85$, and biomass of $W = W_T + W_R$.

We calculate the biomass for cypress by adopting the cypress biomass model with a trunk biomass of $W_S = a(D^2H)^b$, branch biomass of $W_B = c + d(D^2H)$, leaf biomass of $W_L = e + f(D^2H)$, total above-ground biomass of $W_T = W_S + W_B + W_L$, underground biomass of $W_R = g + h(D^2H)$, and biomass of $W = W_T + W_R$, where $a$, $b$, $c$, $d$, $e$, $f$, $g$, and $h$ are model parameters [47].

## 3. Results

### 3.1. Evaluation of Tree Species Identification Based on Multispectral Data

#### 3.1.1. Evaluation of Samples before Classification

For each category of training samples, the transformed divergence method is used to evaluate the separation degree of spectral eigenvectors from different categories. The conversion dispersion is given as $TD_{ij} = 2(1 - e^{-D_{ij}/8})$, where $D_{IJ}$ is the dispersion of two categories and is expressed as:

$$D_{ij} = \frac{1}{2}t_r\left[\left(\textstyle\sum_i - \sum_j\right) \times \left(\textstyle\sum_i^{-1} - \sum_j^{-1}\right)\right] + \frac{1}{2}t_r\left[\left(\textstyle\sum_i^{-1} + \sum_j^{-1}\right) \times (U_i - U_j) \times (U_i - U_j)^T\right] \tag{2}$$

where $U$ is the sample mean vector, $\sum$ is the covariance matrix, $\sum^{-1}$ is the inverse matrix of the covariance matrix, $t_r[A]$ is the sum of the diagonal elements of matrix $A$, and $i$ and

*j* represent the two ground object types, which are valued between 0 and 2. Generally speaking, 1.8 is selected as the threshold value for the sample separation between different objects [48]. When the sample separation is more than 1.8, it is qualified. The greater the separation, the better the computer's ability to distinguish two kinds of ground objects. The transformed divergence results of the samples are shown in Table 6.

**Table 6.** Transformed divergence of samples.

| Training Samples | Poplar | Cypress | Clove Tree | Lacebark Pine |
|---|---|---|---|---|
| Willow | −1.877 | −1.939 | −1.979 | −1.999 |
| Poplar | | −1.999 | −1.945 | −1.999 |
| Cypress | | | −1.999 | −1.998 |
| Clove tree | | | | −1.999 |
| **Validation Samples** | **Poplar** | **Cypress** | **Clove Tree** | **Lacebark Pine** |
| Willow | −1.874 | −1.981 | −1.967 | −1.999 |
| Poplar | | −1.968 | −1.910 | −1.998 |
| Cypress | | | −1.997 | −1.869 |
| Clove tree | | | | −1.999 |

### 3.1.2. Evaluation of Results after Classification

The kappa analysis (kappa coefficient method) adopts a discrete multivariate technique that considers all factors of the matrix. This is an index to measure the coincidence or accuracy between two graphs [49]. The formula is given as:

$$K_{hat} = \frac{N \sum_{i=1}^{r} x_{ii} - \sum_{i=1}^{r} (x_{i+}x_{+i})}{N^2 - \sum_{i=1}^{r} (x_{i+}x_{+i})} \tag{3}$$

where $r$ is the total number of columns in the error matrix; $x_{ii}$ is the number of pixels in row $i$ and column $i$ in the error matrix; $x_{i+}$ and $x_{+i}$ are the total number of pixels in row $i$ and column $i$, respectively; and $N$ is the total number of pixels used.

The overall classification accuracy is the number of correctly classified samples divided by the total number of samples, which only uses the number of pixels located along the diagonal. The $K_{hat}$ considers not only the correctly classified pixels located on the diagonal but also various missing points and misclassification errors not along the diagonal. These two indicators are often inconsistent. In the evaluations, more accurate information is attained by calculating the above indicators [50]. Table 7 displays the results of spectral angle classification.

**Table 7.** Spectral angle classification results.

| Overall Accuracy | (40/78) 51.28% | | Kappa Coefficient | 0.42 | |
|---|---|---|---|---|---|
| **Tree** | **Willow** | **Poplar** | **Cypress** | **Clove Tree** | **Lacebark Pine** |
| Commission (Percent) | 58.33 | 0.00 | 28.57 | 53.85 | 10.00 |
| Omission (Percent) | 79.17 | 28.57 | 50.00 | 50.00 | 18.18 |

### 3.2. Biomass Based on Classified Tree Species

3.2.1. Comparison between DBH and Tree Height

The CCC and RMSE were used to evaluate the relationship between the DBH parameters obtained from the TLS and UAV-LS+TLS and the tree height obtained by UAV-LS and

UAV-LS+TLS. Compared with the Pearson correlation coefficient, the merit of CCC is that it can detect offsets in the measurements and gain offsets. The calculation is given as [31]:

$$CCC = \frac{2\rho\sigma_{12}}{\sigma_1^2 + \sigma_2^2 + (\mu_1 - \mu_2)^2} \tag{4}$$

where $\rho$ denotes the correlation coefficient between two variables and $\sigma^2$ and $\mu$ mean the variance and mean of measures. The RMSE is utilized to measure the magnitude and mean sign difference (MSD) of the modeled volume deviation [31].

(1)　A comparison between the DBH and tree height of various species is shown in Figure 8.

(2)　A comparison between DBH and tree height of all tree species is shown in Figure 9.

### 3.2.2. Biomass Calculations for Each Tree Species

First, tree species were classified through multispectral data combined with field investigations. Then, multispectral classification data were combined with LIDAR point-cloud data to classify point-cloud tree species. Finally, the biomass of each tree species was calculated. Calculations of the tree species biomass can be seen in Table 8.

**Table 8.** Calculation results of tree species biomass.

| Tree Species | Volume/m$^3$ | $W_T$/kg | $W_R$/kg | W |
|---|---|---|---|---|
| Willow | 13.99 | 12,251.55 | 3182.22 | 15,433.77 |
| Poplar | 10.29 | 9105.79 | 2365.14 | 11,470.93 |
| Clove tree | 6.56 | 5738.49 | 1490.52 | 7229.01 |
| Cypress | 12.73 | 11,257.70 | 3881.98 | 15,139.68 |
| Lacebark pine | 1.42 | 1256.47 | 326.36 | 1582.83 |

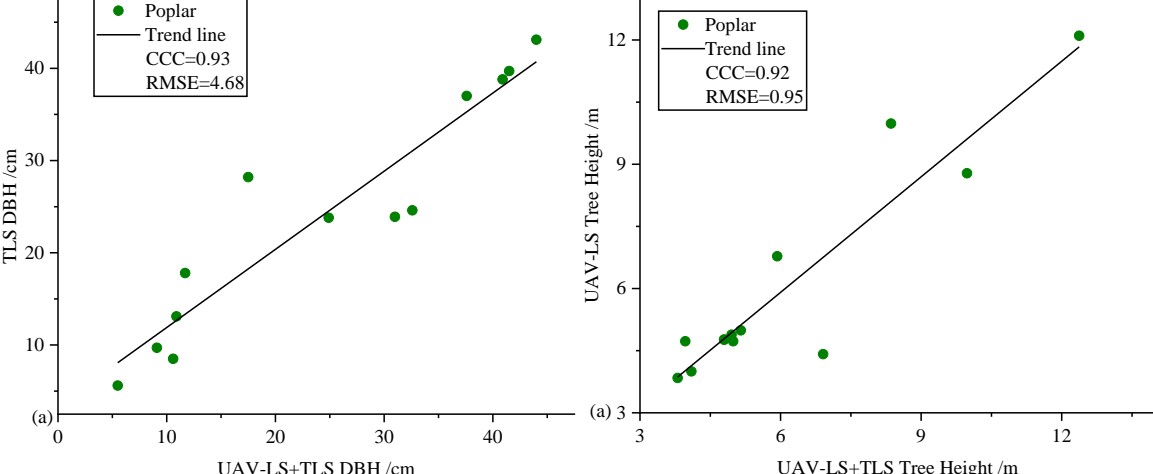

**Figure 8.** *Cont.*

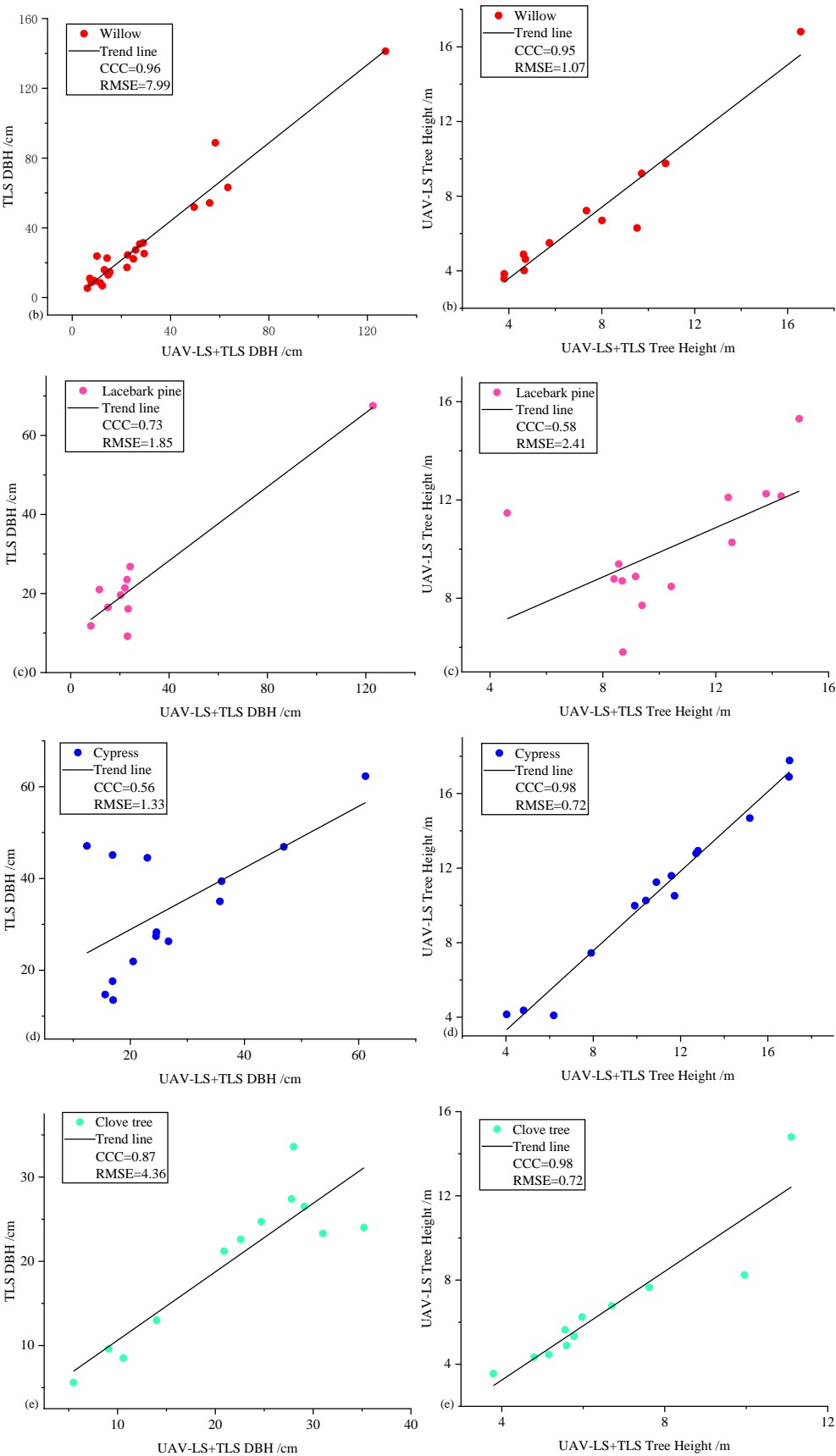

**Figure 8.** The DBH and tree height fit diagrams for each tree species. (**a**) Poplar, (**b**) Willow, (**c**) Lacebark pine, (**d**) Cypress and (**e**) Clove tree.

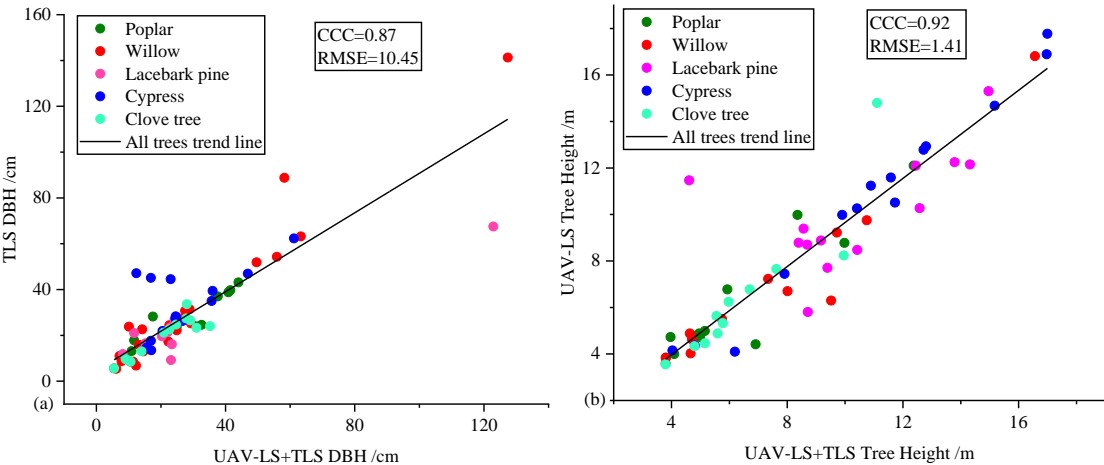

**Figure 9.** (**a**) The DBH fit for all tree species and (**b**) All tree heights.

## 4. Discussion and Conclusions

### 4.1. Multispectral Identification of Tree Species

A UAV equipped with a multispectral sensor can provide an improved resolution. The position and shape of plants can be seen on the image map. The tree species identified were used to extract the spectrum of each plant from the image, and the tree species were classified by building a database. When extracting spectral curves, the various tree species in the stand in the blue, green, and red bands differ slightly, with large differences in the infrared 2 (750 nm) and near-infrared (840 nm) bands. The reflectance values of various tree species within this wavelength range differ greatly, indicating that the infrared and near-infrared bands play important roles in species classification. The same tree species in the study area can have variability in their spectra, and there can be foreign objects with the same spectrum due to differences in the tree species density, age, shadows, and other factors, as well as different tree species, due to the surrounding environment. The reflectivity of the same tree species produces different spectra under different sunlight conditions, and various tree species produce the same spectrum, which impacts the classification accuracy [51–53]. The selection and number of training samples also impact the classification.

According to the field survey and orthophoto map, some knowledge of the study area is used to select training samples to accurately represent the spectral characteristics of each category in the entire region. The training samples of the same category are homogeneous, do not contain other categories, and are not boundaries or mixed pixels between other categories. When the collected spectral curves of various tree species are built into a database for the classification of tree species, the threshold of the spectral angle greatly impacts the classification results. The SAM algorithm compares the unknown spectral lines and classifies them with the sample spectral spectrum in n-dimensional space [42]. When the angles between the reference spectrum vector and each pixel vector are compared, a greater angle makes the pixels more similar to the reference spectrum. Thus, multispectral identification of tree species needs further exploration.

### 4.2. Individual Tree Parameter Segmentation and Comparison

Unlike the forest biomass estimated by optical remote sensing at the stand level, the 3D structural parameters of trees can be obtained from LIDAR point cloud data, but tree species cannot be classified from it. The individual plant biomass is directly associated with the tree size such as DBH and tree height [54–57]. However, due to the influence of site conditions, tree structure, and low vegetation in the study area, the data collected by UAV laser scanning (UAV-LS) are susceptible to canopy influence, and the information below the canopy is missing. Meanwhile, the data collected by terrestrial laser scanning (TLS) are susceptible to canopy influence, and the information above the canopy is missing.

The integrated UAV-LS+TLS greatly increases the accuracy of individual tree parameter extraction and provides a prediction error with reasonable volume. In a single stand, the individual tree parameters can be obtained by LIDAR to determine the biomass. However, in complex stands, obtaining individual tree parameters requires identifying the tree species by optical remote sensing to divide the stand and combine the multispectral tree species classification with LIDAR point cloud data [58–60]. This provides the individual tree biomass for each tree species. The CHM segmentation depends on manually setting the width of the search window to filter the local maximum. As the crowns of willow and poplar are large, each branch may contain more than one local maximum, which often identifies a single tree as several trees. This causes multiple points per tree and artificially increases the number of trees after CHM segmentation. Thus, this process cannot perform automatic and batch processing. It is also necessary to further improve individual plant detection and segmentation based on UAV least squares [61–63].

The CCC of lacebark pine and cypress is relatively low when comparing the parameters of individual trees of each species after segmentation. This is because they are distributed near willows with large tree heights and many small branches, which block the UAV field of view during data collection and result in relatively low point cloud densities. The point cloud density is a critical factor that affects the accurate acquisition of individual tree biomass parameters [31,34,64]. Improving the density of point clouds by modifying the flight path, flying from multiple angles, and selecting an appropriate flight altitude help obtain individual wood parameters efficiently and accurately. Comparing the individual tree parameters in the entire stand shows that the CCC of the DBH is 0.87 and the RMSE is 10.45, while the CCC of the tree height is 0.92 and the RMSE is 1.41. The experimental results show that TLS has a high consistency with the UAV-LS+TLS when the DBH and UAV-LS are obtained for tree heights.

### 4.3. Conclusion

Orthophoto data are used for field investigations of tree species, which are classified in conjunction with multispectral data. The multispectral data and LIDAR point cloud data are combined to classify each tree species. Finally, the individual tree biomass of each species after classification is calculated. The research results show the following.

(1) In the study area, the multispectral tree species identification shows that the extracted spectra can accurately identify Lacebark pine, and the identification of other tree species is slightly lower than that of Lacebark pine. Thus, the reflectance spectrum of Lacebark pine can be applied to the identification of Lacebark pine species.

(2) The comparison of DBH after TLS and UAV-LS+TLS and the comparison of tree height after UAV-LS and UAV-LS+TLS under appropriate conditions show that the DBH parameters obtained by TLS and tree height parameters obtained by UAV-LS have good availability.

(3) In complex stands, multispectral techniques can be used to identify tree species, and LIDAR technology can be used to perform individual tree parameter calculations. Accurately combining the two data, it is feasible to identify tree species in complex forest stands and calculate the biomass of individual trees.

**Author Contributions:** Conceptualization, X.L. and H.Z.; methodology, X.L.; software, H.Z. and Y.L. (Yu Li); validation, W.X., Y.L. (Yunping Lei) and L.G.; formal analysis, X.L.; investigation, Y.H. (Yu Han), H.F., Q.D., L.L., L.S. and J.C.; resources, X.L.; data curation, Y.H. (Yuanwen He); writing—original draft preparation, X.L.; writing—review and editing, X.L.; visualization, X.L.; supervision, K.Q.; project administration, X.L.; funding acquisition, X.L. All authors have read and agreed to the published version of the manuscript.

**Funding:** This research was fund by the National Natural Science Foundation of China [Award Numbers: 42101414 and 51704205] and the Natural Science Foundation of Shanxi Province, China [Grant No. 201901D111074].

**Data Availability Statement:** Not applicable.

**Acknowledgments:** The authors would like to thank the editor and anonymous reviewers for their helpful comments and suggestions.

**Conflicts of Interest:** The authors declare no conflict of interest.

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
