# Peer review of "Biomass Calculations of Individual Trees Based on Unmanned Aerial Vehicle Multispectral Imagery and Laser Scanning Combined with Terrestrial Laser Scanning in Complex Stands"

_remotesensing, doi:10.3390/rs14194715_

Round 1

Reviewer 1 Report

Dear Authors,

kindly find my comments below:

consider rephrasing the title, e.g. "Biomass calculations of individual trees based on unmanned aerial vehicle multispectral imagery and laser scanning combined with terrestrial laser scanning in complex stands"

Section 2.2.2 "In April 2022, the TLS..."

Section 2.2.2 if an ms50 was usaed, I wouldn't call it TLS. Or at least explain how a "real" TLS can improve the dataset by capturing much more points.

Section 2.2.2 "scanning accuracy of 0.6 mm" what is scanning accuracy? Is it 3D accuracy or ranging accuracy - usually the latter is given in the technical specifications of the device.

Section 2.2.2 horizontal interval - what does it mean? How come it is 15m and 10 cm in the same sentence? Consider rephrasing.

Abbreviation of GNSS is explained in Chap 1, no need to do so in Section 2.2.1 and 2.2.2

Section 2.2.3 on what platform was the d-cam2000 mounted? UAV? Anyhow, include it in the text!

Section 2.2.3 GSD is 2cm and not 2cm/pixel

Section 2.2.3 Pix4D is not a Software of "Pegasus UAV", it's not a software dedicated to a device/brand.

Section 2.3.1 "UAVs carrying multispectral technology have made great progress in forestry applications." I expect to emphasize this in the Chapter Introduction with references

Section 2.3.2.1 how werethe point clouds integrated? both georeferenced? It is worth to mention in the text.

Section 2.3.2.2 "Then, the combination of computer algorithms and artificial intelligence generates individual tree seed points." What algorithms and how the AI supports them? A bit more explanation is required here.

Section 3.1.1 "??[?]" r is in subscript

Section 3.1.1 "When greater than 1.8" Why 1.8?

Section 4.1 "A UAV equipped with multispectral equipment" consider: "A UAV equipped with multispectral sensor"

Section 4.1 and the tree species are classified

Section 4.2 I'm missing some kind of summarizing desciption about the differences of the technologies applied; what are the advantages of point clouds and what are the limitations of them? Why UAV LS cannot be used alone, etc.

Conslusion: I would expect stronger statement here: which technologies are to be used for certain purposes, what has been proven from the technological aspects.

Reviewer 2 Report

In this paper, the authors combined orthophoto data and multispectral image data to classify tree species in the study area, and the classified data were combined with LiDAR data to classify tree species from point cloud data, and then the individual biomass of various tree species in complex forest stands was calculated. In this study, the spectral identification of tree species was combined with multi-spectral, UAV-LS, TLS and other multi-source remote sensing data:UAV-LS to obtain tree height and TLS to obtain diameter at breast height. A practical UAV- ls +TLS+multispectral method is proposed to estimate forest biomass. The results show that the combination of UAV-LS and TLS can greatly improve the extraction of individual tree parameters and provide reasonable prediction errors for volume calculation.

This work of the manuscript is innovative with a grace representation of Figure 6. Meanwhile, Multi-source remote sensing fusion would still be of value and has scientific merits, but still has the following shortcomings in readability and need further improvements: 1) In the research method section of 2.3, the method of classifying each tree species by combining the spectral angle classification results with LIDAR point cloud data need to be articulated in detail.

In the introduction section, some work of forest parameter retrieval need to be mentioned, e.g., Shortwave radiation calculation for forest plots using airborne LiDAR data and computer graphics.

In subsection 2.3.2, the strategy of registration UAV-LS and TLS data need to be elaborated, e.g., control points for two data set registration, the utilized software or detrimental effects stemming from occlusion. 

In subsection 2.3.2.2, please elucidate the calculation of the individual tree biomass, using an allometric growth equation or regression equation. How to determine the corresponding coefficients of these equations for different tree species?

The formula in the result section should belong to the method section, symbol sigma represents the summation rather than a matrix.

Reviewer 3 Report

In the forestry research based on UAV remote sensing,  there is not much sepecific research on calculating the individual tree biomass in complex stands. This paper calculateed the individual tree biomass of various tree species in complex stands by combining multispectral and LIDAR data. 

In the first part, the author emphasizes that The digital terrain model data can be subtracted from the digital surface model data to calculate the CHM, and individual trees can be segmented to obtain the individual tree parameters. That's true. But this is sort of contradicting among DTM, DSM and CHM. For the 3D point cloud from Drone, the filtering mothod is a quite crucial process, this is a foundation under the forest application.  

In general, the paper is well-organized and the following papers could be referenced.

Dimitrios Bolkas, Jeffrey Chiampi, John Chapman & Vincent F. Pavill (2020) Creating a virtual reality environment with a fusion of sUAS and TLS point-clouds, International Journal of Image and Data Fusion, 11:2, 136-161, DOI: 10.1080/19479832.2020.1716861

Xiangguo Lin & Wenhan Xie (2022) A segment-based filtering method for mobile laser scanning point cloud, International Journal of Image and Data Fusion, 13:2, 136-154, DOI: 10.1080/19479832.2022.2047801

Reviewer 4 Report

General Comments

The general idea of estimating individual tree biomass based on UAV-LS+TLS+ multispectral method is good, but the manuscript was badly written. I think the following aspects need to be taken into consideration.

1. The study site is a local area of a park in the Huyu campus of Taiyuan University of Technology, which is not a complex stand, but a woodland with five tree species. Thus, the study site was not reasonably selected. In such a small area, 21 survey stations were arranged for obtaining the TLS data, that is not feasible in the application of forest inventory.

2. The authors stated the proposed method was more convenient, efficient, and low-cost than conventional approach, but they had not conducted the field survey through conventional method as the base for comparison. Any RS-based methods would have estimation errors. To evaluate the accuracy and efficiency of the proposed method, we need have a comparison with the results, time and/or cost of field survey.

3. There are only 115 trees of 5 species in total on the study area (called as “complex stand”), and the results in Table 2 show the separability for classifying species is better, but the overall accuracy is only 51% and Kappa Coefficient is only 0.42 (see Table 3), which indicate that the results are very poor.

4. I can not understand why the authors conduct comparison of DBH and tree height values between UAV-LS / TLS and UAV-LS+TLS with CCC and RMSE? From Fig.5, we know that for DBH estimation, UAV-LS almost have no contribution, and for tree height estimation, TLS usually give negative-bias estimation. Combining UAV-LS with TLS, we can utilize advantages and overcome disadvantages of the two methods, and accurately obtain both DBH and tree height estimates. Thus, all comparisons should be conducted viewing from this point.

Specific Comments

The manuscript have not been numbered for every page and line, thus it is not convenient to locate the position of problems. Several specific comments are as follows:

1.     Page 4: On the bottom of the page, the geographical coordinates of the study area is not consistent with the Figure 1;

2.     Page 13: On the column and row of the Table 2, why are there only 4 tree species? (No willow on the row and no cypress on the column)

3.     Page 14: The calculation equation of CCC should have a reference.

4.     Page 14-16: The unit of DBH in Figures 8 and 9 should be cm, not m.

5.     Page 16: Where are the parameters of volume model and biomass models come from? Some references should be added.

Reviewer 5 Report

p.4: What is the area of the observed site? How many trees are surveyed?

p.5: The material was collected in April, does it have an impact on the research of the vegetation or non-vegetation period?

p. 5: Measurement of DBH with GNSS under treetops? That is probably a mistake. If not, how accurate was it? What GNSS equipment was used?

p. 6: The MS50 is not a typical scanner. A product citation is missing.

p. 6 (2.2.2 (1)): What is meant by the sentence? "Twenty-one survey stations were arranged, where the horizontal scanning interval at each station was 15 m, the horizontal interval was 10 cm, and the vertical interval was 10 cm." Are 40,000 points enough, when the laser scanner scanned hundreds of millions of points in 25 min? In which SW were the data processed?

p.6 (2.2.2 (2)): Pegasus d-Lidar - missing product citation, we know nothing about technical parameters. There is no indication of the accuracy achieved after processing. In which SW were the data processed?

p.6 (2.2.3 (1)): Pegasus d-Cam2000 - missing product citation, we know nothing about technical parameters. There is no indication of the accuracy achieved after processing. In which SW were the data processed?

p.6 (2.2.3 (2)): Pegasus d-mspc2000 - missing product citation, we know nothing about technical parameters. There is no indication of the accuracy achieved after processing.

p.7: What does "pix4d Pegasus UAV software" mean? Can be date from Pegasus  in Pix4d SW processed only?

p.7 fig.2 (b): it is a total station with scanning mode, not a scanner. I think the TS details should be more illustrative.

p.7 figure 2 (c): it is a flight plan, not a route plan.

p.11: What is "high-order and low-order" noise?

In general:

How did the co-registration of UAV and TLS scans go?

What SW was used for data classification?

Too complicated (up to 4-level) division of the article.

The conclusion is poorly worded, and says nothing about the scientific contribution; on the contrary, things that are well known enough.

Round 2

Reviewer 2 Report

I have no further concern on the manuscript and agree to publish in current form.

Author Response

Thank you for your kindly review.

Reviewer 4 Report

The authors have improved the manuscript according to the comments of 5 reviewers, and the issues I concerned have been explained in the response. I think it may be accepted for publication after minor revision.

The column for Willow and two rows for Lacebark pine can be deleted in Table 6. The inconsistent order of tree species in row and column in Table 2 of the first version manuscript lead to mistakes of the divergence results.

Reviewer 5 Report

I am satisfied with the modifications made.

Author Response

Thank you for your kindly review.